# GC-IMS-Based Volatile Characteristic Analysis of *Hypsizygus marmoreus* Dried by Different Methods

**DOI:** 10.3390/foods13091322

**Published:** 2024-04-25

**Authors:** Pufu Lai, Longxiang Li, Yingying Wei, Junzheng Sun, Baosha Tang, Yanrong Yang, Junchen Chen, Li Wu

**Affiliations:** 1Institute of Food Science and Technology, Fujian Academy of Agricultural Sciences, Fuzhou 350003, China; laipufu@163.com (P.L.); 13616901013@163.com (L.L.); wyy787459024@163.com (Y.W.); sunjzll@163.com (J.S.); tbsty@126.com (B.T.); 18960973035@163.com (Y.Y.); junchenccc@163.com (J.C.); 2National R & D Center for Edible Fungi Processing, Fuzhou 350003, China; 3Key Laboratory of Subtropical Characteristic Fruits, Vegetables and Edible Fungi Processing (Co-Construction by Ministry and Province), Ministry of Agriculture and Rural Affairs, Fuzhou 350000, China; 4College of Life Sciences, Fujian Agriculture and Forestry University, Fuzhou 350002, China; 5College of Food Science, Fujian Agriculture and Forestry University, Fuzhou 350002, China

**Keywords:** *Hypsizygus marmoreus*, GC-IMS, volatile flavor, drying method

## Abstract

Gas chromatography–ion mobility spectroscopy (GC-IMS) was used to analyze the volatile components in dried *Hypsizygus marmoreus* of different drying methods, including hot air drying (HAD), heat pump drying (HPD), heated freeze-drying (HFD), and unheated freeze-drying (UFD). A total of 116 signal peaks corresponding to 96 volatile compounds were identified, including 25 esters, 24 aldehydes, 23 alcohols, 13 ketones, 10 heterocyclic compounds, 8 carboxylic acids, 7 terpenes, 3 sulfur-containing compounds, 2 nitrogen-containing compounds, and 1 aromatic hydrocarbon. The total content of volatile compounds in *H. marmoreus* dried by the four methods, from highest to lowest, was as follows: HAD, HPD, HFD, and UFD. The main volatile compounds included carboxylic acids, alcohols, esters, and aldehydes. Comparing the peak intensities of volatile compounds in dried *H. marmoreus* using different drying methods, it was found that the synthesis of esters, aldehydes, and terpenes increased under hot drying methods such as HAD and HPD, while the synthesis of compounds containing sulfur and nitrogen increased under freeze-drying methods such as HFD and UFD. Nine common key characteristic flavor compounds of dried *H. marmoreus* were screened using relative odor activity values (ROAV > 1), including ethyl 3-methylbutanoate, acetic acid, 2-methylbutanal, propanal, methyl 2-propenyl sulfate, trimethylamine, 3-octanone, acetaldehide, and thiophene. In the odor description of volatile compounds with ROAV > 0.1, it was found that important flavor components such as trimethylamine, 3-octanone, (*E*)-2-octenal, and dimethyl disulfide are related to the aroma of seafood. Their ROAV order is HFD > UFD > HPD > HAD, indicating that *H. marmoreus* using the HFD method have the strongest seafood flavor. The research findings provide theoretical guidance for selecting drying methods and refining the processing of *H. marmoreus*.

## 1. Introduction

*Hypsizygus marmoreus* (Peck) H. E. Bigelow has a high content of protein, dietary fiber, B vitamins, and minerals, while also exhibiting anti-tumor and antioxidant activities, making it a mushroom that combines both edible and medicinal values [1,2]. *H. marmoreus* is one of the most popular edible fungi in East Asia, and China is the largest producer of cultivated edible fungi [3]. According to statistics from the China Edible Fungi Association, the total production of *H. marmoreus* in China reached 526,300 tons and 546,200 tons in 2021 and 2022, respectively, with year-on-year growth rates of 26.43% and 3.78%, demonstrating significant market potential [4]. The white *Hypsizygus marmoreus* is a strain of *H. marmoreus* that has a taste similar to seafood. Hence it is also known as “seafood mushroom” [5]. Freshly harvested *H. marmoreus* have a high water content, high enzyme activity, and strong respiratory activity, making them susceptible to microbial invasion and rot, resulting in a shelf life of only 3~4 days after harvest [6]. Drying is a widely used method for the long-term preservation of edible mushrooms, and multiple studies have shown that drying significantly alters volatile compounds, thus affecting the flavor [7,8]. Based on the characteristic flavors of dried edible fungi, developing processed products such as dried soup premixes and flavorings is an essential direction for enhancing the product value of *H. marmoreus* and other edible fungi [9].

Currently, flavor research on dried edible fungi has been extensively conducted on common edible fungi such as *Lentinus edodes*, *Ganoderma lucidum,* and *Agaricus bisporus*. *L. edodes* subjected to microwave vacuum drying could better retain flavor-active amino acids, while the content of volatile compounds was significantly increased [10]. The drying process of *G. lucidum* increases the content of aldehydes, esters, and olefins while reducing the content of alcohols and ketones [11]. *A. bisporus*, when freeze-dried, have a reduced content of octenol compounds present compared to those in the fresh mushrooms and generate heat-sensitive alkanes and heterocyclic compounds, which were then degraded during microwave vacuum drying [12]. Different drying methods involved different temperatures and required times, which directly influence the complex reactions related to the production of volatile compounds, such as the Maillard reaction, Strecker degradation, and enzyme-catalyzed reactions [6]. The types and quantities of volatile compounds expressed vary, resulting in differences in flavor profiles. The research system for the post-drying flavor characteristics of edible fungi is quite mature. However, studies on post-drying *H. marmoreus* mainly focus on quality characteristics [13] and non-volatile flavor components [14], with relatively limited research on the impact of drying methods on volatile flavor components.

Gas chromatography–ion mobility spectrometry (GC-IMS) combines the high separation efficiency of gas chromatography with the high sensitivity of ion mobility spectrometry [15]. Its advantages include low detection limits, operation at atmospheric pressure, no need for sample enrichment or concentration, and low cost [16,17]. It has been widely used in the characterization and differentiation of volatile compounds [18,19]. This technique has been employed to identify and analyze the characteristic volatile compounds of three dried *L. edodes* [20]. Additionally, it has been used to establish characteristic volatile fingerprints for both fresh and dried *Tricholoma matsutake* [21], demonstrating its applicability in flavor research of edible fungi. In this study, four common drying methods were used to dehydrate *H. marmoreus*, and the characteristics and differences of volatile compounds after drying were analyzed using GC-IMS. The results help elucidate the mechanisms behind the differences in volatile components and provide references for the development of processed products using *H. marmoreus*.

## 2. Materials and Methods

### 2.1. Materials and Equipment

The fresh *H. marmoreus* used for experimental processing were grown for 120 days under normal conditions and met the commercial requirements. They were purchased from Gutian County, Ningde City, Fujian Province, China. In the laboratory, after removing the bottom substrate of the fresh *H. marmoreus*, individuals with uniform size of fruiting bodies were selected. Their average moisture content was measured using a moisture analyzer (Ohaus Instruments Ltd., Shanghai, China) and found to be 88.20 ± 1.45%. Four portions of *H. marmoreus* weighing 1000 g were dried uniformly with four different drying methods until the moisture content was below 12%. The procedures for the four drying methods are detailed as follows:

Hot air drying (HAD): The drying temperature of the constant temperature blast drying oven was set to 60 °C with an airflow rate of 8 m·s^−1^. After the temperature had stabilized, the samples were laid flat and placed inside, and the drying process lasted for approximately 8 h.

Heat pump drying (HPD): The drying temperature of the heat pump dryer was set to 60 °C with a circulating airflow rate of 2800 m^3^·h^−1^. After the temperature had stabilized, the samples were laid flat and placed inside for a drying duration of approximately 8 h.

Heated freeze-drying (HFD): The samples were spread out on a material tray and pre-frozen at −40 °C for 12 h before being placed in the freeze dryer. The temperature gradient of the freeze dryer was set to −30 °C for 2 h, 30 °C for 8 h, and then stabilized at 60 °C, with a total drying time of approximately 36 h.

Unheated freeze-drying (UFD): The samples were spread out on a material tray and pre-frozen at −40 °C for 12 h before being placed in the freeze-dryer. The temperature of the freeze-dryer was set to 0 °C, and the drying time was approximately 36 h.

The process diagram for drying *H. marmoreus* using four different methods is illustrated in Figure 1. In the previous study, a comprehensive comparison of the physical properties and nutritional quality of *H. marmoreus* dried by the four methods was conducted. The results revealed that all four drying methods were suitable for the industrial production of *H. marmoreus*. UFDHM had the highest content of polysaccharides and polyphenols, HPDHM had the highest total flavonoid content, and the physical characteristics (color, texture, and tissue structure) of the two freeze-dried methods were relatively better [22].

### 2.2. GC-IMS Analysis

The dried *H. marmoreus* (1.0 g) from different drying treatments were placed into glass headspace vials at 60 °C. The vials were incubated at a speed of 500 rpm for 20 min, and then 500 μL of gas was injected into the injector (85 °C, no split mode). The FlavourSpec^®^ flavor analysis instrument (GAS, Dortmund, Germany) was used for GC-IMS measurement, and the retention indices (RI) of each compound were calculated using normal ketones C4~C9 (purchased from China National Pharmaceutical Group Chemical Reagent Co., Ltd., Beijing, China) as external references. Each drying type was sampled three times for the experiment. The instrument procedures and analysis conditions can be found in Appendix A.

### 2.3. Statistical Analysis

The data analysis was performed using the FlavourSpec^®^ system. The VOCal processing software (0.4.03) was used to view the analysis plots, and substance identification was conducted through the NIST database (2020) and IMS database. The spectra differences between samples were directly compared using the Reporter plugin. The fingerprint spectra were compared using the Gallery Plot plugin. Sample clustering analysis was conducted using the Simca software (14.1). The Euclidean distance between each pair of samples was calculated using the Fingerprint Similarity Analysis plugin for Euclidean distance analysis. The relative content of compounds was analyzed for variance and significance (*p* < 0.05) using SPSS software (21.0).

## 3. Results and Discussion

### 3.1. Differences in Volatile Compounds of H. marmoreus Dried by Different Methods

To compare the differences in the volatile flavor compounds of *H. marmoreus* dried using different methods, the Reporter plugin was used to generate 3D GC-IMS spectra of *H. marmoreus* dried by four different methods (Figure 2). From Figure 2, it can be observed that the peak positions of each drying group are roughly the same, indicating that the volatile components of *H. marmoreus* dried by different methods are similar in terms of types. However, there are differences in peak intensities among the groups. Additionally, the ion peak intensities in the highlighted region of Figure 2 are significantly higher than those in the other treatment groups, suggesting that the volatile components corresponding to this region may be specific to the HAD.

To analyze the differences in the volatile components among the treatment groups more intuitively, the 3D GC-IMS spectra were projected into a top-down 2D plot (Figure 3). In Figure 3, most of the ion peaks are located within the retention time range of 0 to 1000 s and the migration time range of 1.0 to 1.5 ms. Additionally, the hot-air-dried group exhibits high concentrations of volatile substances near a retention time of 2000 s and a migration time range of 1.0 to 1.4 ms. This highlighted region in the 2D plot corresponds to the characteristic peak area of the hot-air-dried group in the 3D spectra.

By creating differential spectra from the GC-IMS plots, the differences in the volatile flavor compounds of *H. marmoreus* dried by different methods were visually compared. The GC-IMS plot of HAD was selected as the reference, and the signals from the other drying methods were subtracted to obtain the differential spectra (Figure 4). In Figure 4, most volatile compounds in the HAD reference group had higher concentrations than the other drying groups, indicating that *H. marmoreus* had the highest relative content of volatile compounds under HAD. Additionally, there was a significant difference between HAD and two freeze-drying methods, while the difference between HAD and HPD was relatively small.

### 3.2. Qualitative Analysis of Volatile Compounds in H. marmoreus Dried by Different Methods

A total of 140 signal peaks were detected in *H. marmoreus* dried by four different methods. Based on the retention time and migration time matching with substances in the IMS database, the qualitative analysis resulted in 116 signal peaks corresponding to 96 volatile compounds.

Among the 96 qualitatively identified compounds, they can be classified into different categories. There are 25 esters (including hexyl acetate, pentyl acetate, isoamyl acetate, butyl acetate, ethyl 3-methylbutanoate, and methyl 2-methylbutanoate in both monomeric and dimeric forms), 24 aldehydes (including 2-methyl-2-pentenal, (*Z*)-2-pentenal, and hexanal in both monomeric and dimeric forms), 23 alcohols (including (*E*)-2-hexenol, 1-hexanol, 1-pentanol, 3-methyl-1-butanol, 1-butanol, 1-propanol, and 2-butanol in both monomeric and dimeric forms), 13 ketones (including 1-hydroxy-2-propanone, 3-hydroxy-2-butanone, and 2-heptanone in both monomeric and dimeric forms), 10 heterocyclic compounds, 8 carboxylic acids (including butanoic acid, 2-methylpropanoic acid, propionic acid, and acetic acid in both monomeric and dimeric forms), 7 terpenes (including myrcene in both monomeric and dimeric forms), 3 sulfur compounds, 2 nitrogen compounds, and 1 aromatic hydrocarbon.

The categories and peak intensities of volatile compounds in each group were analyzed, and the total peak intensities and percentage contents of each compound category were obtained, as detailed in Appendix A. Among the four drying methods, HAD had the highest total content of volatile compounds, followed by HPD, HFD, and UFD with the lowest content. Based on the relative content of volatile compounds under different drying methods (Figure 5), it can be observed that the volatile substances in *H. marmoreus* under the four drying methods are mainly carboxylic acids, alcohols, esters, and aldehydes, with their total peak intensities accounting for an over 80% relative proportion. The relative proportions of these four classes of volatile components from largest to smallest are carboxylic acids > alcohols > esters > aldehydes.

### 3.3. The Fingerprints of Volatile Compounds of H. marmoreus Dried by Different Methods

The differences in the volatile components of *H. marmoreus* dried by different drying methods were observed through GC-IMS two-dimensional spectrograms. Due to the difficulty in analyzing closely spaced signal peaks on the spectrograms, the Gallery Plot plugin generated fingerprint spectra of volatile flavor components under four drying methods (Figure 6). By visually and quantitatively comparing the complete volatile component information in the fingerprint spectra, it was observed that the three random replicates of different treated samples exhibited consistency. The peak intensities of all compounds were analyzed for differences under different drying methods, as shown in Table 1.

Carboxylic acids were the predominant volatile compounds in the *H. marmoreus* dried using four different ways, originating from the hydrolysis of fats to short-chain volatile fatty acids or from the degradation of amino acids [23]. Among the eight identified carboxylic acid compounds, the total peak intensity of carboxylic acid compounds under HAD was significantly higher compared to the other three drying methods (*p* < 0.05). In contrast, the total peak intensities under the remaining three drying methods did not differ significantly, indicating that HAD favored the formation of carboxylic acid compounds. Specifically, the peak intensities of butanoic acid and propanoic acid-D were the highest under HAD, while the peak intensities of propanoic acid-M and acetic acid-D were the highest under HPD (*p* < 0.05).

Alcohols are primarily produced by the degradation of unsaturated fatty acids by enzymes such as lipoxygenase and peroxygenase [24], and they are the main volatile compounds in *H. marmoreus* dried by four different methods. The total peak intensity of alcohol compounds ranked highest to lowest among the four drying treatments is as follows: HAD > UFD > HFD > HPD. Among the 23 identified alcohol compounds, the peak intensities of eight compounds (3-methyl-1-butanol-M, 1-butanol-M, 2-methyl-1-propanol, etc.) were the highest under HAD compared to the other three drying methods, while the peak intensities of six compounds (3-methyl-1-pentanol, (*E*)-2-hexenol-M, 2-propanol, etc.) were the highest under HPD. The peak intensities of four compounds (3-methyl-1-butanol-D, 1-butanol-D, ethanol, etc.) were the highest under UFD.

Esters were formed through esterification reactions between carboxylic acids and alcohols [25], and they were the primary volatile compounds in *H. marmoreus* dried by four different methods. The total peak intensity of ester compounds in HAD was similar to that of HPD and significantly higher than the two freeze-drying methods, indicating that both HAD and HPD favored the formation of ester compounds.

Among the 25 identified ester compounds, the peak intensities of nine ester compounds (isoamyl acetate-M, butyl acetate-M, ethyl 3-methylbutanoate-M, etc.) were the highest under HAD compared to the other three drying methods, while the peak intensities of ten ester compounds (isoamyl acetate-D, butyl acetate-D, etc.) were the highest under HPD (*p* < 0.05). Additionally, the peak intensities of three ester compounds (hexyl acetate-M, butyl acetate-M, and ethyl 3-methylbutanoate-D) were significantly higher in both HAD and HPD compared to the two freeze-drying methods (*p* < 0.05).

Aldehydes were products of the lipoxygenase pathway and Strecker degradation [26] and were the main volatile compounds in dried *H. marmoreus*. The total peak intensity of aldehyde compounds under HPD was significantly higher than the other three drying methods (*p* < 0.05). Among the 24 identified aldehyde compounds, the peak intensities of eight compounds (3-methylbutanal, 2-methylpropanal, acetaldehyde, etc.) were highest under HAD compared to the other three drying methods. In HFD, the peak intensities of nine compounds ((*E*)-2-octenal, nonanal, pentanal, etc.) were the highest, while under HPD, only heptanal and (*Z*)-2-pentenal-D showed the highest peak intensities (*p* < 0.05) among aldehyde compounds. Acrolein is a harmful volatile compound commonly found in food, could be generated by the high-temperature processing of fatty-rich foods, and was frequently encountered in baked, fermented, and pickled foods [27]. The World Health Organization’s chemical safety regulations specify a tolerable daily intake of acrolein for the human body at 7.5 µg/kg·bw [28]. However, acrolein’s peak intensity was generally low (426~911) under the four drying methods, and it was difficult to assess its harmful effects on human health. Furthermore, it was only present as a key volatile compound in the HFDHM (Table 2), providing cherry and almond odors [29].

Ketones originated from amino acid degradation, the Maillard reaction, and the thermal oxidation of unsaturated fatty acids [30]. The total peak intensity of ketone compounds was also highest under HAD (*p* < 0.05), with little difference among the other three drying methods, indicating that HAD also had a specific promoting effect on the formation of ketone compounds. Among the 13 identified ketone compounds, eight compounds (2-octanone, 2-butanone, acetone, etc.) had the highest peak intensities under HAD compared to the other three drying methods (*p* < 0.05).

Heterocyclic compounds, sensitive to heat and mainly originating from the Maillard reaction and pyrolysis [31], were detected in dried *H. marmoreus*, including furans, pyrazines, pyridines, and thiophenes. Among the 10 identified heterocyclic compounds, compared to the other three drying methods, HAD had the highest peak intensities for seven compounds (2-methylpyrazine, 2-pentylfuran, 2-butylfuran, etc.), and the total peak intensity of heterocyclic compounds was also highest under HAD (*p* < 0.05).

Terpenes are widely present secondary metabolites in organisms, classified into monoterpenes, sesquiterpenes, diterpenes, triterpenes, and polyterpenes based on the number of carbon atoms they contain [32]. The terpenes detected in dried *H. marmoreus* were all monoterpenes. Among the seven identified terpene compounds, compared to the other three drying methods, HAD had the highest peak intensities for four compounds (alpha-pinene, beta-pinene, etc.), and the total peak intensity of terpenes was also highest under HAD (*p* < 0.05).

Sulfur-containing compounds are commonly present in the volatile components of edible mushrooms. For instance, Dimethyl trisulfide and Dimethyl disulfide were key flavor substances in *L. edodes* after HAD [33]. Compared to the other three drying methods, HFD exhibited the highest peak intensities of dimethyl trisulfide and methyl 2-propenyl sulfide and had the highest total peak intensity of sulfur-containing compounds (*p* < 0.05).

Among nitrogen-containing compounds, the peak intensity of trimethylamine was highest in HFD and UFD. Additionally, the total peak intensity of HFD and HPD was significantly higher than that of HAD and HPD (*p* < 0.05), indicating that the heating–drying process affects the formation of nitrogen-containing compounds.

A discussion can be conducted on the peak intensities of some volatile compounds. The Maillard reaction is a non-enzymatic browning reaction between carbonyl compounds and amino compounds [34], and the quantity of heterocyclic compounds corresponds to the extent of the Maillard reaction [35]. Based on the peak intensities of heterocyclic compounds in different drying methods, it can be observed that HAD exhibited the highest degree of the Maillard reaction, followed by HPD and HFD, while the reaction was significantly inhibited in UFD. Pyrazine compounds are heat-sensitive and are formed through the Maillard reaction and the Strecker degradation of reducing sugars and amino acids [36]. Their peak intensity was highest in HAD (*p* < 0.05), confirming the highest degree of the Maillard reaction in HAD. The initial products of the Maillard reaction are mainly precursors of volatile flavor compounds and browning products [37], and their content is related to the temperature, oxygen, and water activity [38,39]. Therefore, non-vacuum, heated drying conditions determine the Maillard flavor and browning degree of *H. marmoreus*. The high temperature and oxygen content in HAD and HPD contribute to the accumulation of Maillard reaction precursors, and the exhaust efficiency of HAD is lower than that of HPD, resulting in higher heat transfer efficiency in HAD and a higher degree of the Maillard reaction. The temperature in HFD relies on plate conduction, making it difficult to ensure a uniform overall temperature during drying. The lack of oxygen under vacuum conditions weakens lipid oxidation and the generation of carbonyl compounds [40], thus reducing Maillard reaction substrates and weakening the Maillard reaction. Unheated freeze-drying is conducted throughout the process in a vacuum environment at 0 °C or below, inhibiting the progress of the Maillard reaction due to the low temperature and lack of oxygen.

**Table 2 foods-13-01322-t002:** The ROAV of volatile compounds of *H. marmoreus* using four drying methods.

NO.	Compound	Odor Description	Odor Threshold (μg/L)	ROAV			
				HAD	HPD	HFD	UFD
10, 11	Ethyl 3-methylbutanoate	apple, pineapple, fruity	0.00011	100.0000	100.0000	100.0000	100.0000
102, 103	Acetic acid	sour, pungent, vinegar	0.013	12.5090	13.4381	81.4056	41.4407
44	2-Methylbutanal	cocoa, almond	0.001	1.7932	3.5902	37.9957	19.0935
41	Propanal	alcohol, cocoa, nutty	0.0048	3.8091	4.0892	27.7542	13.6690
113	Methyl 2-propenyl sulfide	garlic, onion, alliaceous	0.0005	3.1181	1.9411	26.7787	9.6130
114	Trimethylamine	fishy, pungent	0.02	1.6057	2.5491	24.2904	13.9812
78	3-Octanone	herbal, fresh, mushroom	0.0013	2.3167	2.0636	21.2537	14.2491
43	Acetaldehyde	whiskey, pungent, fruity	0.0027	2.2393	1.8539	13.8660	7.0853
98, 99	2-Methylpropanoic acid	butter, strawberry, cheese	0.04	1.9117	0.7959	12.8856	7.7170
95	Thiophene	garlic, alliaceous	0.0019	2.0711	1.3325	11.1635	7.3497
40	Butanal	chocolate, herbaceous, floral, fruity	0.002	0.4692	0.4184	5.6678	1.3346
29	Heptanal	citrus, fatty, rancid	0.003	0.3988	0.5889	3.9985	1.4950
27	(*E*)-2-Octenal	nuts, green, fatty	0.003	0.1910	0.3891	3.5683	1.2711
64	2-Methyl-1-propanol	solvent, ether, wine, bitter	0.033	0.4660	0.3599	2.6215	1.7375
92	2-Pentylfuran	green beans, vegetable	0.006	0.6026	0.4423	2.3472	1.3056
90	Dimethyl disulfide	vegetable, nutty, meaty, green	0.0084	0.1463	0.3206	2.1086	1.2490
39	Acrolein	cherry, almond	0.0083	0.1952	0.1764	2.0832	0.5202
91	2-Butylfuran	wine, sweet, fruity, spicy	0.005	0.4275	0.3574	1.9303	1.3321
42	2-Methylpropanal	malt, pungent, green	0.0015	1.2649	0.6996	1.6304	0.6312
65, 66	1-Propanol	fermented, fusel, pungent	0.24	0.1407	0.1072	0.7133	0.4071
84	3-Nonanone	jasmin, herbal, fresh	0.017	0.0284	0.0633	0.5642	0.3300
45	cis-4-Heptenal	biscuit, dairy, green	0.040	0.0043	0.0154	0.4871	0.3523
62, 63	1-Butanol	vanilla, fruit, balsam	0.48	0.0322	0.0267	0.2572	0.1513
19	Ethyl Acetate	pineapple, anise, fruity, green	0.88	0.0343	0.0326	0.2430	0.1493
69	Ethanol	ethereal, sweet	0.62	0.0304	0.0300	0.2279	0.1478
8, 9	Butyl acetate	sweet, banana	0.13	0.0374	0.0410	0.2262	0.1452
37	Pentanal	bready, berry, almond	0.4	0.0168	0.0208	0.1783	0.0593
6, 7	Isoamyl acetate	banana, fruity, sweet	0.918	0.0337	0.0382	0.1616	0.1130
60, 61	3-Methyl-1-butanol	sweet, malty, rubber	1.69	0.0220	0.0204	0.1523	0.0928
83	Acetone	apple, pear, ethereal	0.832	0.0262	0.0157	0.1043	0.0467

Note: The content of the monomer and dimer of the same substance is calculated after adding them up. HAD: Hot air drying; HPD: Heat pump drying; HFD: Heat freeze-drying; UFD: Unheated freeze-drying. The odor thresholds are from “Compilations of odour threshold values in air, water and other media (second enlarged and revised edition)” [41] and “Odor thresholds for chemicals with established occupational health standards (second edition)” [42].

### 3.4. ROAV Analysis of Volatile Compounds in H. marmoreus Dried by Different Methods

The relative odor activity value (ROAV) is widely used to characterize the contribution of volatile flavor compounds to the overall flavor [43]. Compounds with a ROAV value which is not less than 1 are typically defined as key flavor compounds, while those with a ROAV value between 0.1 and 1 are considered to have a modifying effect on the overall flavor [44]. The ROAV value was calculated based on the relative contents of each compound in *H. marmoreus* obtained through four different drying methods, as shown in Table 2. From Table 2, it could be observed that the number of compounds contributing to flavor varies from highest to lowest among the four drying methods as follows: HFD, UFD, HAD, and HPD, which also corresponds to the order of the number of key flavor compounds. Additionally, all four drying methods contain 20 common compounds contributing to flavor (ROAV > 0.1), among which nine compounds were identified as key flavor compounds (ROAV > 1) across all four drying methods. These nine common key flavor compounds were ethyl 3-methylbutanoate, acetic acid, 2-methylbutanal, propanal, methyl 2-propenyl sulfide, trimethylamine, 3-octanone, acetaldehyde, and thiophene. Their aroma characteristics collectively form the basic aroma of dried *H. marmoreus*, with an overall aroma profile of fruity, sour, cocoa, garlic, fishy, and mushroom. In addition, according to the composition of *H. marmoreus’* flavor substances dried by four drying methods, there were significant differences in the composition of the flavor-contributing compounds and the ROAV values of identical compounds of *H. marmoreus* between two heat-drying methods (HAD and HPD) and two freeze-drying methods (HFD and UFD). These indicate that the flavor characteristics of the two heat-drying and two freeze-drying methods also exhibited significant differences. The composition of the flavor-contributing compounds in HFD and UFD was essentially the same. The difference lay in the fact that two key flavor compounds (acrolein, 2-methylpropanal) of HFDHM only played a modifying role in the flavor of UFDHM. In comparison, UFDHM has three more compounds (pentanal, 3-methyl-1-butanol, acetone) that play a role in the modifying flavor. Moreover, overall, the ROAV values of the same flavor compounds in HFDHM were generally higher than those in UFDHM, indicating that the aroma in HFDHM was stronger than in UFDHM. Similarly, the flavor characteristics between HADHM and HPDHM are similar. The composition of the flavor compounds contributing to the whole flavor in both were identical, and their ROAV values were close. Among them, 2-methylpropanoic acid and 2-methylpropanal were identified as key flavor compounds in HADHM (ROAV > 1), while in HPDHM, they only played a modifying role in their flavor (0.1 < ROAV < 1). Consequently, the buttery, cheesy, and pungent odors in HADHM were more pronounced than HPDHM’s.

The seafood-like aroma of *H. marmoreus* is a well-known characteristic flavor among the public [45]. This seafood-like aroma corresponds to odor descriptions such as fishy, fresh, and green, typically found in fresh seafood products [46]. In the four methods of dried *H. marmoreus*, flavor compounds related to the odor descriptions included trimethylamine, 3-octanone, (*E*)-2-octenal, and dimethyl disulfide. Among them, trimethylamine is commonly found in various seafood products and is a crucial indicator of seafood freshness [47]. 3-Octanone is identified as a primary flavor compound in fresh *H. marmoreus* [48]. (*E*)-2-octenal is a lipid-derived volatile aroma compound produced by the lipoxygenase/hydroperoxide lyase (LOX/HPL) pathway, which has been shown to contribute to the characteristic flavor of fresh seafood [46,49]. Dimethyl disulfide is a sulfur-containing volatile compound that has been shown to enhance the aroma of some fresh seafood [46]. The relative odor activity values (ROAV) of these seafood-like compounds in the *H. marmoreus* treated with four drying methods roughly followed this sequence: HFD > UFD > HPD > HAD. Consequently, it could be concluded that *H. marmoreus* dried by the HFD method exhibited the most potent seafood-like flavor among the four drying methods.

### 3.5. Cluster Analysis of Volatile Compounds in H. marmoreus Dried by Different Methods

A Principal Component Analysis (PCA) was performed on the volatile compounds of dried *H. marmoreus* using different drying methods (Figure 7). The cumulative contribution rates of the first and second principal components were 79.8%, indicating that PC1 and PC2 in the figure effectively characterized the differences in volatile compounds among the different treatment groups. From Figure 7, it can be observed that the parallel samples of the four drying treatments clustered distinctly, and the distance between HFD and UFD was close, suggesting that the differences in volatile compounds between these two freeze-drying treatments were relatively small, and they might exhibit similar overall flavor characteristics. Additionally, there was a clear separation trend between the two principal components for HAD, HPD, and the two freeze-dried groups, indicating significant differences in volatile compounds.

The “nearest neighbor” fingerprint analysis was used to calculate the Euclidean distance between each pair of treatment groups and retrieve the minimum distance to determine the similarity level of the treatment groups (Figure 8). Figure 8 shows that the parallel samples of the four drying methods clustered distinctly, with the non-heat freeze-dried group exhibiting the most concentrated normal distribution. This indicated that the volatile compounds expressed in *H. marmoreus* under UFD were the most uniform, and the consistency of multiple treatments was better. At the same time, the distribution of the Euclidean distances indicated differences in the composition of volatile compounds among the four drying methods, with a smaller difference between HFD and UFD, and a larger difference between them and the two freeze-drying methods (HAD and HPD), consistent with the conclusion in Figure 7.

## 4. Conclusions

This study analyzed the differences in the volatile compounds of *H. marmoreus* under different drying methods using GC-IMS technology. A total of 116 signal peaks were identified by GC-IMS, including 25 esters, 24 aldehydes, 23 alcohols, 13 ketones, 10 heterocyclic compounds, 8 carboxylic acids, 7 terpenes, 3 sulfur-containing compounds, 2 nitrogen-containing compounds, and 1 aromatic hydrocarbon. The major volatile compound compositions were the same among the four drying methods, with carboxylic acids > alcohols > esters > aldehydes in decreasing order of relative proportions, all accounting for over 80% of the total relative abundance. Significant differences were observed in the peak intensities of volatile compounds under different drying methods, as indicated by the fingerprint patterns and peak intensities. The total peak intensity of alcohols, ketones, and carboxylic acids was highest in HADHM, while that of aldehydes was highest in HPDHM, of sulfur-containing compounds in HFDHM, esters in the two heat-dried methods, and nitrogen-containing compounds in the two freeze-dried methods. According to the results of the relative odor activity values, all four drying methods contained 20 compounds contributing to flavor (ROAV > 0.1), with 9 compounds making a critical contribution to flavor (ROAV > 1). The odor descriptions of these nine key flavor compounds constituted the basic flavor of dried *H. marmoreus*, including fruity, sour, cocoa, garlic, fishy, and mushroom flavors. Furthermore, among their key flavor compounds, trimethylamine, 3-octanone, (*E*)-2-octenal, and dimethyl disulfide have a seafood flavor or enhance the seafood flavor, and the order of the ROAV values for these four compounds was HFD > UFD > HPD > HAD. Therefore, HFDHM had the most potent seafood flavor.

This study revealed the composition characteristics and flavor profiles of *H. marmoreus* dried by four different methods. This study’s results can provide references for the flavor requirements in the fine processing of *H. marmoreus* using various drying methods.

## Figures and Tables

**Figure 1 foods-13-01322-f001:**
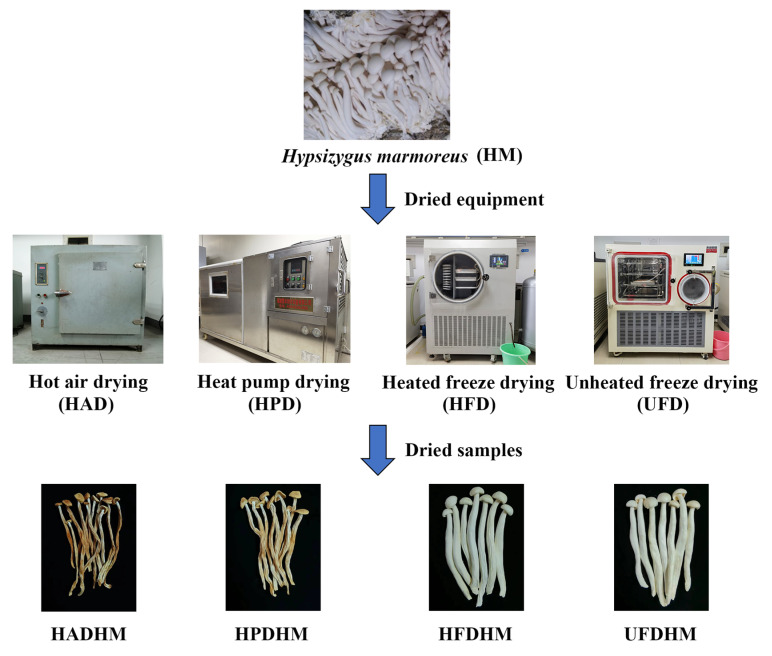
Schematic diagram of different drying methods for processing *H. marmoreus*.

**Figure 2 foods-13-01322-f002:**
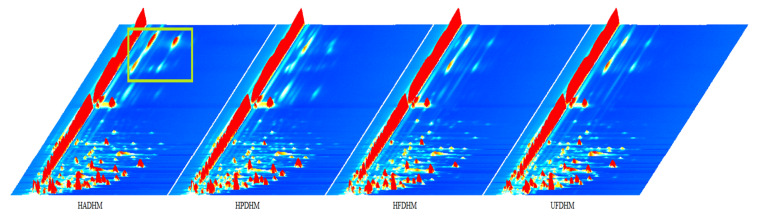
Three-dimensional chromatographic analysis plot of volatile components in dried *H. marmoreus* using four different methods. HADHM: Hot-air-dried *H. marmoreus*; HPDHM: Heat-pump-dried *H. marmoreus*; HFDHM: Heated freeze−dried *H. marmoreus*; UFDHM: Unheated freeze-dried *H. marmoreus*. Migration time, retention time, and peak intensity correspond to the X-axis, Y-axis, and Z-axis in the figure, respectively. The color of each volatile component represents its concentration, with white indicating low concentration and red indicating high concentration. The darker the color, the higher the concentration. The green boxes represent the highlighted region where differences are evident.

**Figure 3 foods-13-01322-f003:**
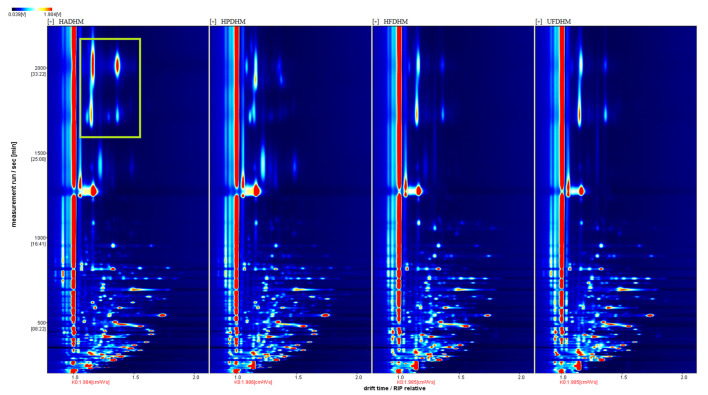
Top-down analysis plot of volatile components in dried *H. marmoreus* using four different methods. HADHM: Hot-air-dried *H. marmoreus*; HPDHM: Heat-pump-dried *H. marmoreus*; HFDHM: Heated freeze-dried *H. marmoreus*; UFDHM: Unheated freeze-dried *H. marmoreus*. The red vertical line at 1.0 on the x-axis represents the normalized response ion peak. The color of each volatile component represents its concentration, with white indicating low concentration and red indicating high concentration. The darker the color, the higher the concentration. The green boxes represent the highlighted region where differences are evident.

**Figure 4 foods-13-01322-f004:**
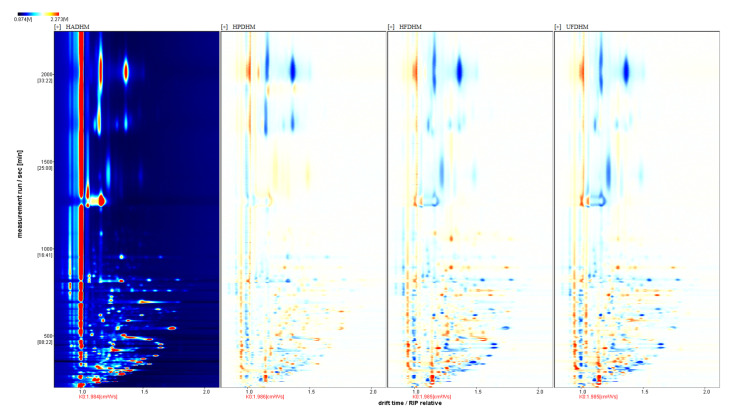
Differential analysis plot of volatile components in dried *H. marmoreus* using four different methods. HADHM: Hot-air-dried *H. marmoreus*; HPDHM: Heat-pump-dried *H. marmoreus*; HFDHM: Heated freeze-dried *H. marmoreus*; UFDHM: Unheated freeze-dried *H. marmoreus*. The blue color represents a lower concentration of the substance compared with the reference, while the red color represents a higher concentration. The darker the color, the more significant the difference.

**Figure 5 foods-13-01322-f005:**
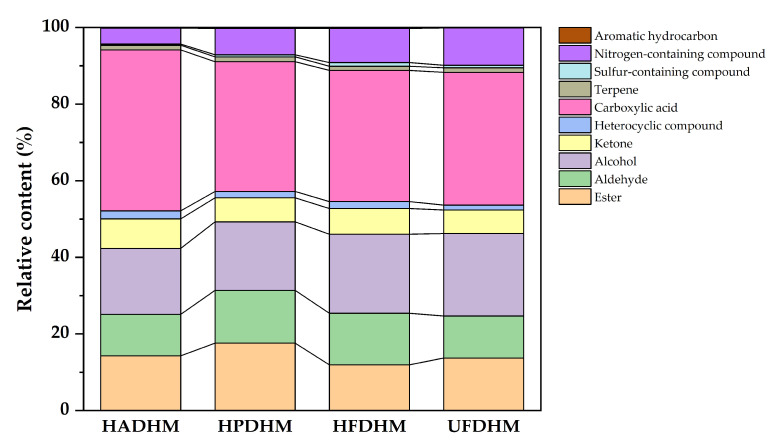
The relative content of volatile compounds of dried *H. marmoreus* using four different methods. HADHM: Hot-air-dried *H. marmoreus*; HPDHM: Heat-pump-dried *H. marmoreus*; HFDHM: Heated freeze-dried *H. marmoreus*; UFDHM: Unheated freeze-dried *H. marmoreus*. Regarding the invisibility of the brown portion, it is because the Aromatic hydrocarbons, has a very low percentage in each group, approximately around 0.2% (see Appendix A).

**Figure 6 foods-13-01322-f006:**
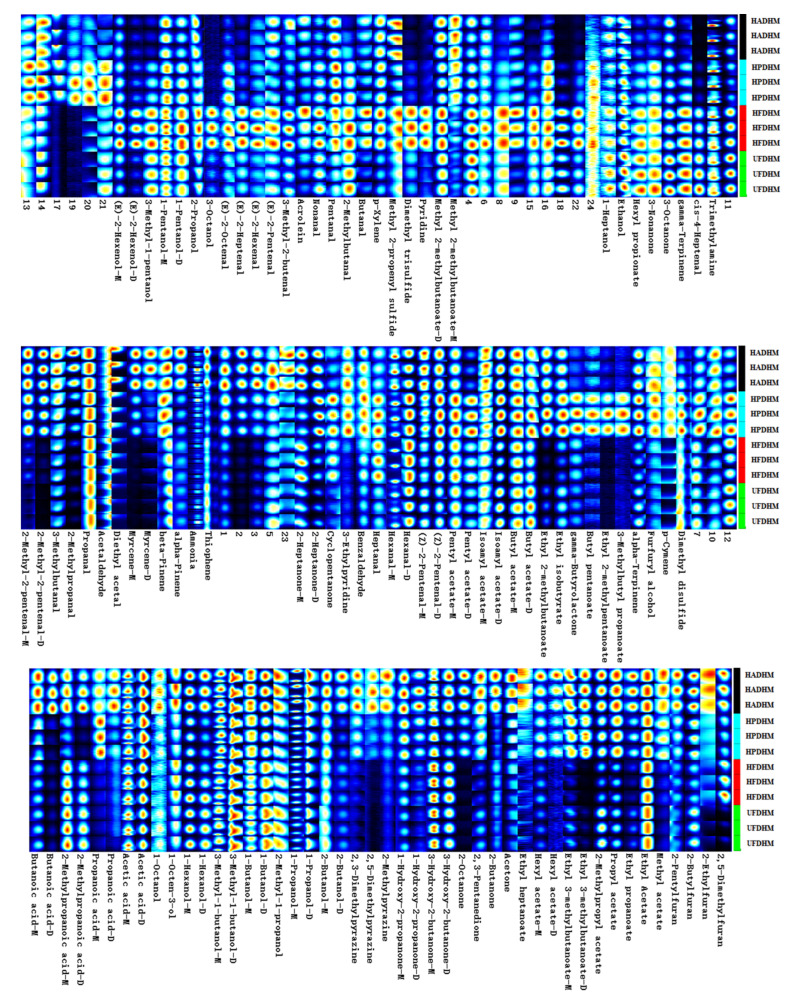
The fingerprint profiles of volatile flavors of dried *H. marmoreus* using four different methods. HADHM: Hot-air-dried *H. marmoreus*; HPDHM: Heat-pump-dried *H. marmoreus*; HFDHM: Heated freeze-dried *H. marmoreus*; UFDHM: Unheated freeze-dried *H. marmoreus*. Each row represents all the peaks of volatile compounds expressed in a sample, while each column represents the expression of the same compound in different samples. The darker the color, the higher the concentration of that compound. Compounds that have not been identified are represented by numbers.

**Figure 7 foods-13-01322-f007:**
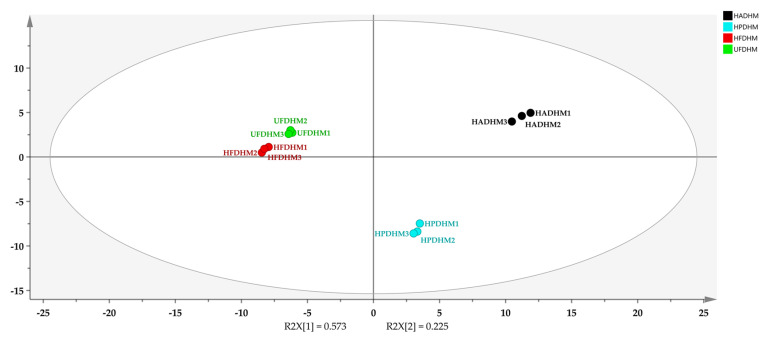
PCA analysis chart of volatile compounds in *H. marmoreus* dried by different methods. HADHM: Hot-air-dried *H. marmoreus*; HPDHM: Heat-pump-dried *H. marmoreus*; HFDHM: Heated freeze-dried *H. marmoreus*; UFDHM: Unheated freeze-dried *H. marmoreus*.

**Figure 8 foods-13-01322-f008:**
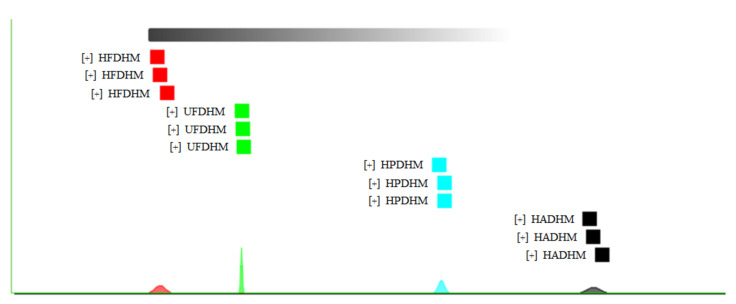
Euclidean distance chart of volatile compounds in *H. marmoreus* dried by different methods. HADHM: Hot-air-dried *H. marmoreus*; HPDHM: Heat-pump-dried *H. marmoreus*; HFDHM: Heated freeze-dried *H. marmoreus*; UFDHM: Unheated freeze-dried *H. marmoreus*.

**Table 1 foods-13-01322-t001:** The volatile compounds and peak intensities of dried *H. marmoreus* using different methods.

NO.	Volatile Compounds	RI	Rt (s)	Dt (RIP Rel)	Peak Intensity			
					HAD	HPD	HFD	UFD
	Esters							
1	Hexyl propionate	1337.7	893.617	1.43424	241.61 ± 10.46 ^d^	464.64 ± 23.18 ^c^	757.47 ± 34.34 ^b^	828.52 ± 28.27 ^a^
2	Hexyl acetate-M	1282.4	793.596	1.3862	674.92 ± 99.27 ^a^	562.1 ± 48.77 ^a^	305.08 ± 29.93 ^b^	267.38 ± 4.71 ^b^
3	Hexyl acetate-D	1282.4	793.596	1.89535	156.32 ± 14.17 ^a^	113.69 ± 16.74 ^b^	60.86 ± 11.3 ^c^	57.72 ± 7.64 ^c^
4	Pentyl acetate-M	1185.3	643.622	1.31204	1150.78 ± 32.37 ^c^	1588.67 ± 12.9 ^a^	1413.03 ± 42.98 ^b^	1221.99 ± 19.45 ^c^
5	Pentyl acetate-D	1185.6	644.29	1.7582	451.8 ± 3.36 ^d^	1351.58 ± 44.56 ^a^	937.06 ± 85.42 ^b^	750.87 ± 3.7 ^c^
6	Isoamyl acetate-M	1135	544.741	1.30162	3365.42 ± 79.76 ^a^	2640.67 ± 133.59 ^b^	2286.55 ± 43.5 ^c^	2317.59 ± 15.86 ^c^
7	Isoamyl acetate-D	1135	544.741	1.74952	9328.41 ± 157.54 ^b^	11,216.06 ± 505.59 ^a^	5546.66 ± 277.33 ^d^	7923.42 ± 70.32 ^c^
8	Butyl acetate-M	1085.7	464.399	1.23694	1067.03 ± 51.34 ^a^	1059.32 ± 13.84 ^a^	873.34 ± 5.62 ^b^	935.29 ± 14.14 ^b^
9	Butyl acetate-D	1085.4	463.895	1.61559	926.87 ± 35.44 ^b^	1048.73 ± 77.18 ^a^	679.65 ± 26 ^c^	927.68 ± 5.24 ^b^
10	Ethyl 3-methylbutanoate-M	1078	453.832	1.26208	2174.67 ± 81.19 ^a^	1995 ± 24.28 ^b^	479.4 ± 45.81 ^d^	833.28 ± 10.68 ^c^
11	Ethyl 3-methylbutanoate-D	1079.1	455.342	1.64961	2336.09 ± 69.4 ^a^	2353.17 ± 178.56 ^a^	100.61 ± 21.15 ^b^	252.28 ± 3.87 ^b^
12	Ethyl 2-methylbutanoate	1063.8	435.215	1.64517	2387.04 ± 81.78 ^b^	2686.19 ± 211.72 ^a^	252.08 ± 32.34 ^d^	633.72 ± 10.77 ^c^
13	2-Methylpropyl acetate	1025	387.918	1.60672	2759.58 ± 170.79 ^a^	1949.85 ± 151.1 ^b^	953.58 ± 74.75 ^c^	1954.3 ± 31.99 ^b^
14	Methyl 2-methylbutanoate-M	1008.9	369.804	1.21475	1014.19 ± 1.84 ^a^	943.87 ± 6.35 ^b^	499.46 ± 7.26 ^c^	364.92 ± 10.69 ^d^
15	Methyl 2-methylbutanoate-D	1009.4	370.307	1.51205	3729.79 ± 230.78 ^b^	2994.94 ± 78.08 ^c^	5011.2 ± 72.91 ^a^	3891.53 ± 102.22 ^b^
16	Propyl acetate	989.4	350.684	1.4736	8152.93 ± 87.35 ^a^	6641.38 ± 227.45 ^b^	2376.46 ± 106.77 ^d^	2844.48 ± 80.86 ^c^
17	Ethyl isobutyrate	976.5	340.62	1.55643	1752.69 ± 34.7 ^b^	2406.14 ± 105.73 ^a^	774.62 ± 171.94 ^d^	1260.66 ± 50.77 ^c^
18	Ethyl propanoate	969.9	335.589	1.44549	1387.63 ± 55.61 ^a^	764.79 ± 42.32 ^b^	297.81 ± 45.09 ^d^	458.48 ± 30.1 ^c^
19	Ethyl Acetate	893.9	282.756	1.3316	12,380.05 ± 281.02 ^a^	11,332.59 ± 110.14 ^b^	11,284.8 ± 557.25 ^b^	12,969.61 ± 252.01 ^a^
20	Methyl acetate	850.8	256.592	1.19848	516.21 ± 41.65 ^a^	302.11 ± 12.99 ^b^	125.11 ± 10.76 ^c^	99.69 ± 9.06 ^c^
21	gamma-Butyrolactone	1708.4	1997.64	1.08832	1430.92 ± 73.16 ^d^	3436.07 ± 160.6 ^a^	2306.96 ± 62.02 ^b^	1829.2 ± 45.2 ^c^
22	Butyl pentanoate	1299.9	823.328	1.92958	171.18 ± 3.44 ^b^	269.15 ± 20.15 ^a^	110.96 ± 2.68 ^c^	113.67 ± 4.78 ^c^
23	Ethyl 2-methylpentanoate	1150.1	572.698	1.76476	91.77 ± 2.56 ^b^	324.18 ± 50.62 ^a^	39.63 ± 1.93 ^b^	32.46 ± 3.24 ^b^
24	3-Methylbutyl propanoate	1184.3	641.448	1.82184	38.84 ± 2.29 ^b^	142.58 ± 10.09 ^a^	32.21 ± 4.77 ^b^	28.79 ± 1.3 ^b^
25	Ethyl heptanoate	1358.9	935.774	1.4221	62.42 ± 3.17 ^a^	28.88 ± 2.28 ^b^	22.28 ± 2.36 ^c^	24.07 ± 1.09 ^bc^
	Total			25 kinds	57,749.18 ± 1645.53 ^a^	58,616.34 ± 1823.31 ^a^	37,526.87 ± 1142.76 ^c^	42,821.6 ± 487.72 ^b^
	Aldehydes							
26	Benzaldehyde	1549.2	1414.329	1.15653	1222.35 ± 66.3 ^a^	1284.04 ± 74.14 ^a^	904.32 ± 108.94 ^b^	819.43 ± 43.81 ^b^
27	(*E*)-2-Octenal	1437.1	1108.822	1.33821	234.96 ± 11.23 ^d^	461.45 ± 21.76 ^b^	565.4 ± 33.75 ^a^	376.31 ± 19.07 ^c^
28	Nonanal	1400.6	1024.411	1.48034	257.34 ± 11.74 ^d^	370.74 ± 7.38 ^c^	528.49 ± 5.25 ^a^	435.71 ± 9.72 ^b^
29	Heptanal	1196.3	663.967	1.34493	490.65 ± 19.48 ^c^	698.53 ± 16.59 ^a^	633.62 ± 27.5 ^b^	442.62 ± 2.3 ^d^
30	2-Methyl-2-pentenal-M	1174.8	621.574	1.161	2138.4 ± 78.52 ^a^	1755.37 ± 27.9 ^b^	672.46 ± 62.04 ^c^	552.39 ± 11.21 ^c^
31	2-Methyl-2-pentenal-D	1174.8	621.574	1.49779	1612.1 ± 68.76 ^a^	1309.62 ± 34.02 ^b^	237.07 ± 14.46 ^c^	77.07 ± 1.34 ^d^
32	(*E*)-2-Pentenal	1144.5	562.112	1.36933	260.2 ± 3.76 ^b^	228.02 ± 3.95 ^c^	332.19 ± 6.15 ^a^	222.08 ± 10.58 ^c^
33	(*Z*)-2-Pentenal-M	1113.6	507.327	1.09329	2131.07 ± 53.72 ^a^	1937.64 ± 25.22 ^b^	1281.12 ± 11.41 ^c^	1296.05 ± 17.96 ^c^
34	(*Z*)-2-Pentenal-D	1113.6	507.327	1.35197	6444.13 ± 243.92 ^b^	7690.42 ± 89.31 ^a^	5870.24 ± 215.59 ^c^	4373.53 ± 148.63 ^d^
35	Hexanal-M	1097.3	480.602	1.27211	2102.39 ± 65.3 ^a^	1637.01 ± 38.43 ^b^	1086.82 ± 38.32 ^d^	1393.17 ± 14.53 ^c^
36	Hexanal-D	1098.1	481.938	1.56029	5269.69 ± 202.28 ^d^	7606.58 ± 168.82 ^b^	8467.13 ± 43.39 ^a^	6108.9 ± 97.85 ^c^
37	Pentanal	999.1	359.237	1.41739	2756.34 ± 98.98 ^c^	3283.39 ± 88.24 ^b^	3764.36 ± 22.79 ^a^	2339.38 ± 37.51 ^d^
38	3-Methylbutanal	925.9	303.889	1.39964	5410.07 ± 242.58 ^a^	4268.28 ± 219.01 ^b^	3056.66 ± 101.49 ^c^	2597.13 ± 49.48 ^d^
39	Acrolein	862.8	263.636	1.05797	664.72 ± 91.54 ^ab^	580.02 ± 100.57 ^b^	910.87 ± 177.78 ^a^	426.15 ± 64.22 ^b^
40	Butanal	882.7	275.712	1.27688	384.91 ± 43.37 ^b^	331.1 ± 17.49 ^bc^	598.2 ± 50.91 ^a^	263.42 ± 8.6 ^c^
41	Propanal	813.6	235.962	1.12453	7497.44 ± 135.36 ^a^	7762.17 ± 12.21 ^a^	7030.3 ± 178.14 ^b^	6474.92 ± 26.1 ^c^
42	2-Methylpropanal	824.8	242.0	1.27983	778.23 ± 68.42 ^a^	414.99 ± 5.78 ^b^	128.65 ± 32.17 ^c^	93.45 ± 17.79 ^c^
43	Acetaldehyde	763.5	210.804	0.97958	2479.01 ± 25.98 ^a^	1979.76 ± 33.2 ^b^	1975.78 ± 49.29 ^b^	1887.97 ± 75.73 ^b^
44	2-Methylbutanal	910.1	293.323	1.18961	734.52 ± 215.47 ^c^	1419.16 ± 69.62 ^b^	2006.24 ± 105.61 ^a^	1884.19 ± 81.01 ^a^
45	cis-4-Heptenal	1245.6	735.435	1.61815	70.84 ± 3.19 ^c^	242.82 ± 15.21 ^c^	1029.64 ± 83.71 ^b^	1390.49 ± 122 ^a^
46	3-Methyl-2-butenal	1213	687.373	1.09618	157.6 ± 25.22 ^ab^	133.03 ± 6.81 ^b^	177.42 ± 4.69 ^a^	135.57 ± 6.21 ^b^
47	(*E*)-2-Hexenal	1230	711.938	1.18348	154.8 ± 0.22 ^b^	170.25 ± 6.16 ^b^	286.31 ± 14.44 ^a^	153.66 ± 2.98 ^b^
48	Diethyl acetal	903.2	288.762	1.02714	276.12 ± 6.25 ^a^	170.46 ± 8.5 ^b^	126.97 ± 8.16 ^c^	74.28 ± 4.67 ^d^
49	(*E*)-2-Heptenal	1327.4	874.016	1.2559	144.33 ± 4.9 ^d^	232.34 ± 8.48 ^c^	659.91 ± 43.22 ^a^	414.24 ± 6.62 ^b^
	Total			24 kinds	43,672.23 ± 1224.98 ^b^	45,967.2 ± 562.19 ^a^	42,330.17 ± 255.35 ^b^	34,232.11 ± 266.53 ^c^
	Alcohols							
50	1-Octanol	1653.7	1774.275	1.48139	738.87 ± 20.36 ^a^	689.76 ± 36.41 ^ab^	477.77 ± 54.1 ^c^	600.69 ± 34.89 ^b^
51	1-Octen-3-ol	1482.7	1224.076	1.16471	975.22 ± 48.6 ^a^	653.88 ± 6.77 ^c^	857.5 ± 10.26 ^b^	690.69 ± 6.14 ^c^
52	(*E*)-2-Hexenol-M	1428.2	1087.72	1.16656	1926.68 ± 221.16 ^c^	2379.84 ± 21.3 ^b^	3381.28 ± 95.24 ^a^	1952.71 ± 147.46 ^c^
53	(*E*)-2-Hexenol-D	1428.2	1087.72	1.51725	199.71 ± 5.13 ^c^	278.09 ± 14.91 ^b^	706 ± 53.34 ^a^	230.34 ± 14.03 ^bc^
54	1-Heptanol	1485.1	1230.57	1.40651	245.94 ± 25.34 ^a^	213.71 ± 7.15 ^a^	230.92 ± 8.56 ^a^	254.91 ± 19.76 ^a^
55	1-Hexanol-M	1367.6	953.629	1.33177	3793.23 ± 105.92 ^a^	2530.37 ± 40.78 ^c^	3100.05 ± 345.88 ^b^	3370.52 ± 95.04 ^ab^
56	1-Hexanol-D	1367.6	953.629	1.64238	813.54 ± 51.17 ^a^	379.91 ± 9.37 ^c^	615.39 ± 140.35 ^b^	727.75 ± 25.9 ^bc^
57	3-Methyl-1-pentanol	1337.7	893.617	1.30135	414.7 ± 22.04 ^d^	1110.91 ± 27.76 ^c^	2023.41 ± 78.93 ^a^	1795.84 ± 59.67 ^b^
58	1-Pentanol-M	1263.4	763.001	1.25492	2197.89 ± 192.52 ^b^	2657.8 ± 37.39 ^a^	2813.86 ± 55.17 ^a^	2582.88 ± 82.92 ^a^
59	1-Pentanol-D	1263.6	763.273	1.71603	902.58 ± 96.53 ^d^	1264.25 ± 40.14 ^c^	1847.54 ± 90.82 ^a^	1631.14 ± 68.75 ^b^
60	3-Methyl-1-butanol-M	1219.2	696.198	1.24061	4449.66 ± 105.95 ^a^	4045.96 ± 41.58 ^b^	3601.74 ± 24.13 ^c^	3587.98 ± 40.99 ^c^
61	3-Methyl-1-butanol-D	1219.8	697.069	1.48891	10,788.77 ± 254.76 ^b^	9589.59 ± 129.5 ^c^	9987.41 ± 226.35 ^c^	11,891.83 ± 56.3 ^a^
62	1-Butanol-M	1157.8	587.501	1.18357	3102.04 ± 51.56 ^a^	2593.62 ± 6.2 ^b^	2463.16 ± 26.15 ^c^	2407 ± 19.52 ^c^
63	1-Butanol-D	1158.5	588.837	1.37974	3233.01 ± 70.61 ^c^	2470.59 ± 29.61 ^d^	4053.32 ± 53.54 ^b^	4760.83 ± 13.58 ^a^
64	2-Methyl-1-propanol	1104.7	492.628	1.39016	6306.28 ± 171.31 ^a^	4695.26 ± 109.49 ^c^	4567.56 ± 138.74 ^c^	5658.44 ± 24.48 ^b^
65	1-Propanol-M	1050.7	418.611	1.11122	1882.13 ± 50.35 ^a^	1627.25 ± 64.08 ^b^	1536.6 ± 25.95 ^b^	1546.95 ± 36.87 ^b^
66	1-Propanol-D	1051.1	419.114	1.24729	11,961.7 ± 478.32 ^a^	8549.22 ± 93.45 ^b^	7500.83 ± 61.2 ^c^	8094.03 ± 79.96 ^bc^
67	2-Butanol-M	1035.8	400.497	1.14967	990.75 ± 49.48 ^b^	952.05 ± 46.01 ^b^	972.38 ± 28.65 ^b^	1165.87 ± 39.56 ^a^
68	2-Butanol-D	1036.2	401	1.32569	640.94 ± 70.13 ^a^	234 ± 8.35 ^b^	299.43 ± 21.73 ^b^	285.32 ± 2.58 ^b^
69	Ethanol	943.2	315.965	1.13488	7727.46 ± 255.36 ^b^	7364.08 ± 131.31 ^b^	7454.53 ± 325.25 ^b^	9043.37 ± 69.68 ^a^
70	2-Propanol	943.2	315.965	1.23694	4257.82 ± 27.13 ^b^	3479.58 ± 14.36 ^d^	4822.08 ± 144.74 ^a^	3735.1 ± 69.94 ^c^
71	Furfuryl alcohol	1733.3	2108.897	1.12724	1893.82 ± 134.66 ^a^	1881.74 ± 43.66 ^a^	1184.24 ± 49.39 ^b^	1064.05 ± 19.02 ^b^
72	3-Octanol	1414.4	1055.589	1.77946	104.85 ± 7.25 ^c^	118.03 ± 5.15 ^c^	452.65 ± 9.74 ^a^	175.51 ± 1.6 ^b^
	Total			23 kinds	69,547.58 ± 1584.08 ^a^	59,759.48 ± 398.29 ^d^	64,949.65 ± 688.18 ^c^	67,253.74 ± 339.13 ^b^
	Ketones							
73	1-Hydroxy-2-propanone-M	1313.4	847.725	1.06919	2426.74 ± 18.5 ^a^	1949.15 ± 12.6 ^b^	1463.61 ± 52.52 ^c^	1238.03 ± 44.05 ^d^
74	1-Hydroxy-2-propanone-D	1313.4	847.725	1.2325	520.19 ± 67.7 ^a^	394.24 ± 18.01 ^b^	240.12 ± 9.12 ^c^	220.14 ± 3.81 ^c^
75	3-Hydroxy-2-butanone-M	1297.1	818.307	1.06759	3504.65 ± 167.54 ^b^	3185.82 ± 48.14 ^c^	3943.9 ± 10.62 ^a^	3767.79 ± 53.03 ^a^
76	3-Hydroxy-2-butanone-D	1297.1	818.307	1.33017	5157.27 ± 232.7 ^a^	2542.07 ± 64.24 ^d^	4253.49 ± 284.87 ^b^	3684.5 ± 67.95 ^c^
77	2-Octanone	1300.4	824.191	1.78488	1083.03 ± 76.4 ^a^	666.59 ± 35.04 ^b^	134.72 ± 4.25 ^c^	137.84 ± 2.82 ^c^
78	3-Octanone	1263.4	763.001	1.71603	1235.36 ± 127.28 ^bc^	1059.89 ± 104.44 ^c^	1458.15 ± 74.36 ^b^	1828 ± 46.49 ^a^
79	2-Heptanone-M	1191.5	656.984	1.26343	1357.14 ± 34.32 ^a^	727.85 ± 20.58 ^d^	1184.67 ± 28.83 ^b^	929.75 ± 31.56 ^c^
80	2-Heptanone-D	1191.5	656.984	1.63321	2237.69 ± 100.71 ^a^	2277.34 ± 111.55 ^a^	1129.18 ± 39.51 ^b^	867.47 ± 53.61 ^c^
81	2,3-Pentanedione	1065	436.725	1.25025	1387.07 ± 96.13 ^a^	958.8 ± 18.06 ^b^	377.64 ± 9.02 ^d^	561.82 ± 11.62 ^c^
82	2-Butanone	911.7	294.329	1.2399	3117.95 ± 186.47 ^a^	1261.81 ± 11.48 ^c^	1696.4 ± 67.78 ^b^	1364.74 ± 10.33 ^c^
83	Acetone	835.7	248.038	1.11269	8956.62 ± 783.61 ^a^	5174.84 ± 149.74 ^b^	4568.71 ± 810.31 ^b^	3831.34 ± 395.24 ^b^
84	3-Nonanone	1337.4	893.138	1.39632	197.65 ± 7.06 ^d^	425.63 ± 19.61 ^c^	506.26 ± 10.34 ^b^	553.58 ± 30.51 ^a^
85	Cyclopentanone	1147.8	568.305	1.10548	156.54 ± 2.94 ^c^	279.27 ± 13.43 ^a^	184.54 ± 13.71 ^b^	133.7 ± 9.83 ^c^
	Total			13 kinds	31,337.91 ± 1695.68 ^a^	20,903.29 ± 81.83 ^b^	21,141.37 ± 683.66 ^b^	19,118.7 ± 406.56 ^b^
	Heterocyclic compounds							
86	3-Ethylpyridine	1386.5	993.568	1.10565	494.71 ± 9.31 ^a^	508.7 ± 18.96 ^a^	206.13 ± 23.53 ^b^	212.88 ± 1.73 ^b^
87	2,3-Dimethylpyrazine	1349	915.974	1.11082	555.23 ± 28.29 ^a^	412.82 ± 14.69 ^b^	196.94 ± 3.82 ^c^	177.47 ± 4.54 ^c^
88	2,5-Dimethylpyrazine	1325.4	870.082	1.12042	438.9 ± 4.51 ^a^	172.45 ± 7 ^b^	63.12 ± 7.48 ^c^	57.8 ± 2.26 ^c^
89	2-Methylpyrazine	1275.2	781.829	1.09641	453.58 ± 23.77 ^a^	214.37 ± 2.85 ^b^	147.87 ± 7.17 ^c^	134.66 ± 4.4 ^c^
90	2-Pentylfuran	1241	728.429	1.2509	1482.58 ± 54.72 ^a^	1049.54 ± 24.67 ^b^	743.91 ± 30.97 ^c^	773.04 ± 6.68 ^c^
91	2-Butylfuran	1107.6	497.305	1.17142	876.46 ± 5.58 ^a^	706.88 ± 19.12 ^b^	509.37 ± 7.29 ^d^	657.32 ± 16.49 ^c^
92	2,5-Dimethylfuran	946.7	318.481	1.37745	2174.36 ± 141.44 ^a^	1152.76 ± 62.46 ^b^	1978.58 ± 307.44 ^a^	317.52 ± 25.71 ^c^
93	2-Ethylfuran	958.4	327.035	1.28871	252.83 ± 22.01 ^a^	74.43 ± 5.81 ^c^	117.04 ± 3.49 ^b^	61.61 ± 2.85 ^c^
94	Pyridine	1170.9	613.68	1.2635	112.19 ± 18.39 ^d^	150.43 ± 4.62 ^c^	616.37 ± 2.79 ^a^	216.79 ± 1.95 ^b^
95	Thiophene	1027	390.207	1.0475	1613.63 ± 39.07 ^a^	1001.38 ± 17.84 ^d^	1118.64 ± 79.38 ^c^	1378.14 ± 29.08 ^b^
	Total			10 kinds	8454.45 ± 313.69 ^a^	5443.76 ± 82.71 ^b^	5697.97 ± 228.32 ^b^	3987.23 ± 37.23 ^c^
	Carboxylic acids							
96	Butanoic acid-M	1712.9	2017.481	1.16247	37,635.9 ± 732.85 ^a^	20,863.3 ± 2142.89 ^b^	18,271.38 ± 519.5 ^b^	18,186.25 ± 147.07 ^b^
97	Butanoic acid-D	1712.4	2015.05	1.36848	28,868.24 ± 1837.27 ^a^	52,63.79 ± 1187.17 ^b^	4516.87 ± 242.04 ^b^	4668.38 ± 72.59 ^b^
98	2-Methylpropanoic acid-M	1640.3	1723.202	1.15059	23,690.86 ± 384.85 ^a^	11,082.69 ± 884.89 ^c^	21,625.23 ± 915.42 ^b^	23,702.71 ± 281.66 ^a^
99	2-Methylpropanoic acid-D	1640.9	1725.634	1.37245	7659.53 ± 811.85 ^a^	1517.3 ± 250.71 ^c^	5605.63 ± 474.18 ^b^	6759.93 ± 141.04 ^ab^
100	Propanoic acid-M	1633.7	1698.881	1.11889	3317.51 ± 51.9 ^b^	4065.17 ± 57.08 ^a^	1145.18 ± 20.61 ^c^	967.87 ± 39.12 ^d^
101	Propanoic acid-D	1635	1703.745	1.29519	1833 ± 104.99 ^a^	1075.68 ± 95.15 ^b^	749.73 ± 14 ^c^	623.17 ± 114.39 ^c^
102	Acetic acid-M	1500.8	1273.27	1.05947	30,454.85 ± 730.64 ^a^	29,370.65 ± 488.98 ^a^	28,918.44 ± 1048 ^a^	29,322.27 ± 185.48 ^a^
103	Acetic acid-D	1500.8	1273.27	1.15653	36,224.97 ± 190.56 ^b^	39,720.15 ± 185.77 ^a^	26,958.14 ± 367.99 ^c^	23,843.66 ± 830.55 ^d^
	Total			8 kinds	169,684.85 ± 3386.03 ^a^	112,958.72 ± 5135.23 ^b^	107,790.6 ± 3056.05 ^b^	108,074.25 ± 964.34 ^b^
	Terpenes							
104	gamma-Terpinene	1246.7	737.14	1.22592	471.25 ± 14.58 ^d^	785.15 ± 22.78 ^c^	1505.13 ± 48.29 ^b^	1666.59 ± 35.72 ^a^
105	alpha-Terpinene	1189.3	652.308	1.22697	1356.99 ± 15.96 ^b^	1484.76 ± 28.85 ^a^	831.85 ± 41.68 ^d^	1031.31 ± 57.85 ^c^
106	Myrcene-M	1180.6	633.6	1.22697	1075.64 ± 21.45 ^a^	541.83 ± 5.87 ^b^	115.37 ± 8.11 ^c^	90.54 ± 1.19 ^c^
107	Myrcene-D	1180.6	633.6	1.64015	326.92 ± 9.47 ^a^	144.96 ± 7.73 ^b^	65.26 ± 1.36 ^c^	44.37 ± 0.7 ^d^
108	beta-Pinene	1144.8	562.78	1.21655	623.32 ± 14.21 ^a^	542.98 ± 22.48 ^b^	382.47 ± 7.39 ^d^	422.19 ± 5.49 ^c^
109	alpha-Pinene	1026.8	389.93	1.28723	610.47 ± 41.58 ^a^	322.49 ± 5.34 ^b^	301.2 ± 5.51 ^b^	287.75 ± 3.08 ^b^
110	p-Cymene	1302.5	827.905	1.29549	388.01 ± 16.68 ^a^	427.03 ± 30.2 ^a^	161.29 ± 6.63 ^b^	138.19 ± 4.61 ^b^
	Total			7 kinds	4852.6 ± 122.64 ^a^	4249.2 ± 65.77 ^b^	3362.57 ± 70.83 ^d^	3680.95 ± 89.49 ^c^
	Sulfur-containing compounds							
111	Dimethyl trisulfide	1412.9	1052.007	1.3013	183.99 ± 9.1 ^d^	396.61 ± 6.65 ^c^	1385.08 ± 25.62 ^a^	590.74 ± 26.53 ^b^
112	Dimethyl disulfide	1048.8	416.246	1.15241	503.64 ± 100.66 ^b^	1064.68 ± 24.48 ^a^	935.1 ± 49.2 ^a^	1035.39 ± 35.01 ^a^
113	Methyl 2-propenyl sulfide	982.4	345.181	1.03654	639.33 ± 20.79 ^b^	383.92 ± 11.65 ^d^	706.6 ± 15.61 ^a^	474.33 ± 7.24 ^c^
	Total			3 kinds	1326.96 ± 88.41 ^d^	1845.21 ± 15.2 ^c^	3026.78 ± 64.68 ^a^	2100.46 ± 19.72 ^b^
	Nitrogen-containing compounds							
114	Trimethylamine	837.5	249.044	1.14524	13,163.86 ± 1177.72 ^c^	20,155.58 ± 621.09 ^b^	25,675.48 ± 2361.68 ^a^	27,593.84 ± 1417.5 ^a^
115	Ammonia	1260.2	757.916	0.85109	3525.72 ± 635.6 ^a^	2840.95 ± 260.16 ^ab^	2313.13 ± 105.74 ^b^	2771.16 ± 205.98 ^ab^
	Total			2 kinds	16,689.58 ± 1745.14 ^c^	22,996.53 ± 371.18 ^b^	27,988.61 ± 2256.53 ^a^	30,365 ± 1623.45 ^a^
	Aromatic hydrocarbons							
116	p-Xylene	1144.5	562.112	1.07593	760.66 ± 23.35 ^b^	719.13 ± 15.16 ^b^	850.45 ± 52.77 ^a^	419.44 ± 14.21 ^c^
	Total			1 kind	760.66 ± 23.35 ^b^	719.13 ± 15.16 ^b^	850.45 ± 52.77 ^a^	419.44 ± 14.21 ^c^

Note: HAD: Hot air drying; HPD: Heat pump drying; HFD: Heat freeze-drying; UFD: Unheated freeze-drying. The suffix “-M” after the compound name indicates that the compound is monomeric, while “-D” indicates that the compound is dimeric. Different letters within the same row indicate significant differences (*p* < 0.05).

## Data Availability

The original contributions presented in the study are included in the article/Appendix A, further inquiries can be directed to the corresponding author.

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
