# Peer review of "GC-IMS-Based Volatile Characteristic Analysis of Hypsizygus marmoreus Dried by Different Methods"

_foods, 2024, doi:10.3390/foods13091322_

Round 1
Reviewer 1 Report
Comments and Suggestions for Authors
In this paper, four common drying methods were used to dehydrate H. marmoreus, and the characteristics and differences of volatile compounds after drying were analyzed using GC-IMS.
The paper is well written, fluent reading.
INTRODUCTION: It would be convenient to add information about the consumption of this type of mushrooms, the worldwide demand, since it is related to the importance of the study of this edible mushroom.
The authors characterize all the VOCs and calculate the aromatics indexes.
The tables and figures are correct.
I suggest adding photos of how the mushrooms look after drying, for each particular drying method, as well as fresh, since the visual aspect is a very important factor for the consumer. As well as including an analysis of product acceptability.
Author Response
Reviewer 1
In this paper, four common drying methods were used to dehydrate H. marmoreus, and the characteristics and differences of volatile compounds after drying were analyzed using GC-IMS. The paper is well written, fluent reading. The authors characterize all the VOCs and calculate the aromatics indexes. The tables and figures are correct.
Q1. INTRODUCTION: It would be convenient to add information about the consumption of this type of mushrooms, the worldwide demand, since it is related to the importance of the study of this edible mushroom.
Response: Thanks for the reviewer’s comment. It is true as the reviewer pointed out that we should add more information about the consumption of H. marmoreus in the Introduction section to show the importance of this research. We have revised the introduction section of the manuscript accordingly. “H. marmoreus is one of the most popular edible fungi in East Asia, and China is the largest producer of cultivated edible fungi [3]. According to statistics from the China Edible Fungi Association, the total production of H. marmoreus in China reached 526,300 tons and 546,200 tons in 2021 and 2022, respectively, with year-on-year growth rates of 26.43% and 3.78%, demonstrating significant market potential [4].” has been added in the Introduction section.
References:
[3] Wu, Y., Shang, J., Li, Y., Zhou, C., Hou, D., Li, J., et al. The complete mitochondrial genome of the Basidiomycete edible fungus Hypsizygus marmoreus. Mitochondrial DNA Part B. 2018, 3, 1241-1243. doi:10.1080/23802359.2018.1532343.
[4] Cao, Z., Chen, H., Zhou, C., Gong, M., Li, Y., Shao, Y., et al. Exogenous γ-aminobutyric acid (GABA) enhanced response to abiotic stress in Hypsizygus marmoreus by improving mycelial growth and antioxidant capacity. Metabolites. 2024. 14(2), 94. doi:10.3390/metabo14020094.
Q2. I suggest adding photos of how the mushrooms look after drying, for each particular drying method, as well as fresh, since the visual aspect is a very important factor for the consumer. As well as including an analysis of product acceptability.
Response: Thank the reviewers for their comments. We have made corresponding revisions to the original manuscript. We have added a process diagram (Figure 1) illustrating the drying methods for H. marmoreus. Additionally, "In the previous study, a comprehensive comparison of the physical properties and nutritional quality of H. marmoreus dried by the four methods was conducted. The results revealed that all four drying methods were suitable for the industrial production of H. marmoreus.UFDHM had the highest content of polysaccharides and polyphenols, HPDHM had the highest total flavonoid content, and the physical characteristics (color, texture, tissue structure) of the two freeze-dried methods were relatively better [22]." has been included as an acceptability analysis for drying H. marmoreus in Section 2.1.
Figure 1. Schematic diagram of different drying methods for processing H. marmoreus.
References:
[22] Lai, P., Tang, B., Li, Y., Wu, L., Weng, M., & Chen, J. Grey correlation analysis for physical and nutritional quality of Hypsizygus marmoreus from different drying methods. Journal of Nuclear Agricultural Sciences. 2021, 35(09), 2118-2126. doi:10.11869/j.issn.100-8551.2021.09.2118.

Reviewer 2 Report
Comments and Suggestions for Authors
The paper was written according to the instructions for the authors. Experimental data are well presented, and tables and figures describe all established data.
In the discussion, I suggest the authors to describe the presence of unacceptable volatile compounds such as acrolein and to discuss the amount of acrolein present in the analyzed samples.
The manuscript describes the Gas Chromatography Mobility Spectroscopy (GC-IMS) method used for the analysis of volatile components in dried Hypsizygus marmoreus by different drying methods, including hot air drying (HAD), heat pump drying (HPD), heated freeze drying (HFD) and unheated freeze drying (UFD). The data was collected in highly scientific recommendations and with all the latest research.
The conclusions are consistent and in accordance with all the collected data.
References are up to date.
I recommend the authors to do minor revision.
Author Response
Reviewer 2
The paper was written according to the instructions for the authors. Experimental data are well presented, and tables and figures describe all established data. In the discussion, I suggest the authors to describe the presence of unacceptable volatile compounds such as acrolein and to discuss the amount of acrolein present in the analyzed samples. The manuscript describes the Gas Chromatography Mobility Spectroscopy (GC-IMS) method used for the analysis of volatile components in dried Hypsizygus marmoreus by different drying methods, including hot air drying (HAD), heat pump drying (HPD), heated freeze drying (HFD) and unheated freeze drying (UFD). The data was collected in highly scientific recommendations and with all the latest research. The conclusions are consistent and in accordance with all the collected data. References are up to date. I recommend the authors to do minor revision.
Q1. In the discussion, I suggest the authors to describe the presence of unacceptable volatile compounds such as acrolein and to discuss the amount of acrolein present in the analyzed samples.
Response: Thanks for the reviewer’s comment. We apologize for overlooking the discussion on the unacceptable volatile compound acrolein. We have now supplemented the relevant content in the article. "Acrolein is a harmful volatile compound commonly found in food, could be generated by high-temperature processing of fatty-rich foods, and it was frequently encountered in baked, fermented, and pickled foods [27]. The World Health Organization's chemical safety regulations specify a tolerable daily intake of acrolein for the human body at 7.5 µg/kgï½¥bw [28]. However, acrolein's peak intensity was generally low (426 ~ 911) under four drying methods, and it was difficult to assess its harmful effects on human health. Furthermore, it was only present as a key volatile compound in the HFDHM (Table 2), providing cherry and almond odors [29]." has been added to Section 3.3.
References:
[27] Liu, J., Lu, Y., Si, B., Tong, A., Lu, Y., & Lv, L. Synergistic Inhibitory Effect of Multiple Polyphenols from Spice on Acrolein during High-Temperature Processing. Foods. 2023, 12, 2326. doi:10.3390/foods12122326.
[28] Gomes, R., & Meek, M. World Health Organization & international Programme on Chemical Safety. World Health Organization: Geneva, Switzerland. 2002.
[29] Li, B., Zhang, T., Dai, Y., Jiang, G., Peng, Y., Wang, J., et al. Effects of probiotics on antioxidant activity, flavor compounds and sensory evaluation of Rosa roxburghii Tratt. LWT-Food Science and Technology. 2023, 179, 114664. doi:10.1016/j.lwt.2023.114664.

Round 2
Reviewer 1 Report
Comments and Suggestions for Authors
Suggestions were considered by the authors. The paper is ready to be accepted